# Smooth Quadratic Prediction Markets

**Enrique Nueve**
Department of Computer Science
University of Colorado Boulder
enrique.nueveiv@colorado.edu

**Bo Waggoner**
Department of Computer Science
University of Colorado Boulder
bwag@colorado.edu

## Abstract

When agents trade in a Duality-based Cost Function prediction market, they collectively implement the learning algorithm Follow-The-Regularized-Leader [Abernethy et al., 2013]. We ask whether other learning algorithms could be used to inspire the design of prediction markets. By decomposing and modifying the Duality-based Cost Function Market Maker's (DCFMM) pricing mechanism, we propose a new prediction market, called the Smooth Quadratic Prediction Market, the incentivizes agents to collectively implement general steepest gradient descent. Relative to the DCFMM, the Smooth Quadratic Prediction Market has a better worst-case monetary loss for AD securities while preserving axiom guarantees such as the existence of instantaneous price, information incorporation, expressiveness, no arbitrage, and a form of incentive compatibility. To motivate the application of the Smooth Quadratic Prediction Market, we independently examine agents' trading behavior under two realistic constraints: bounded budgets and buy-only securities. Finally, we provide an introductory analysis of an approach to facilitate adaptive liquidity using the Smooth Quadratic Prediction Market. Our results suggest future designs where the price update rule is separate from the fee structure, yet guarantees are preserved.

## 1 Introduction

**What are Prediction Markets?** Prediction markets allow traders to buy and sell securities whose payoffs depend on the realization of future events [Hanson, 2003]. Prediction market platforms such as Kalshi and Polymarket on average move millions of dollars per event-based market. In this work, we examine prediction markets that trade Arrow-Debreu (AD) securities, which pay \$1 if a particular state of the world is reached and \$0 otherwise [Arrow, 1964]. Given a finite random variable $\mathcal{Y} = \{y_1, \ldots, y_d\}$ over $d$ mutually exclusive and exhaustive outcomes, AD securities are designed to elicit a full probability distribution with respect to $\mathcal{Y}$, reflected by the current prices of the securities.

**Duality-based CFMM** The work of Abernethy et al. [2013] introduced a general framework for designing automated prediction markets over combinatorial or infinite state outcome spaces named the Duality-based Cost Function Market Maker (DCFMM). As stated by Abernethy et al. [2013], "An automated market maker is a market institution that adaptively sets prices for each security and is always willing to accept trades at these prices." The DCFMM satisfies the desirable market axioms of no arbitrage, bounded worst-case loss, information incorporation, expressiveness, incentive compatibility, and efficient computability. The DCFMM over the last decade has been examined in numerous works such as Abernethy et al. [2014], Devanur et al. [2015], Frongillo and Waggoner [2017], Frongillo et al. [2023]. A core observation of Abernethy et al. [2013] is that the DCFMM uses Follow-The-Regularized-Leader (FTRL) to set prices for securities, by interpreting past trades as loss vectors. Although not our main contribution, as it has been hinted at in the literature before by Devanur et al. [2015], to our knowledge, we are the first to formally show (in Appendix C) a tight

39th Conference on Neural Information Processing Systems (NeurIPS 2025).

equivalence between Continuous FTRL and DCFMM in terms of regret and worst-case monetary loss of the mechanism. Inspired by this relation, we ask if other machine learning algorithms could be used to design a new prediction market with at least the same or better properties than the DCFMM. This work proposes a new market maker payment mechanism called the Smooth Quadratic Prediction Market, which is analogous to the learning algorithm general steepest descent in terms of trader's incentives.

**Smooth Quadratic Prediction Markets** We propose the Smooth Quadratic Prediction Market, which increases profitability for the market maker relative to the DCFMM but still has the performance guarantees of the DCFMM, such as no arbitrage, bounded worst-case loss, information incorporation, expressiveness, computational efficiency, and a form of incentive compatibility. We show that the Smooth Quadratic Market can be interpreted as replacing a Bregman divergence term in the DCFMM payments by a simpler quadratic "fee". In the body of this work, we show that the Smooth Quadratic Prediction Market has many of the same axiom guarantees as that of the DCFMM, demonstrate how agents are incentivized to follow general steepest descent when trading, examine how agents trade under either bounded budgets or a buy-only market, and provide an approach to facilitate adaptive liquidity.

## 2 Background and Notation

Let us denote the all ones vector by $\mathbb{1} = (1, \ldots, 1) \in \mathbb{R}^d$. Let us denote by the vector $\delta_i = (0, \ldots, 1, \ldots, 0)$, i.e. all-zero except for a one in the $i$-th position. Comparison between vectors is pointwise, e.g. $q \succ q'$ if $q_i > q_i'$ for all $i = 1, \ldots, d$ and similarly for $\succeq$. We say $q \gneq q'$ when $q_i \geq q_i'$ for all $i$ and $q \neq q'$. We denote $((u)_+)_i = \max(u_i, 0)$. For a set $Y \subseteq \mathbb{R}^d$, we denote by closure$(Y)$ as the smallest closed set containing all of the limit points of $Y$. Define $\mathbb{R}_+^d$ to be the nonnegative orthant. Let $\Delta_d = \{p \in \mathbb{R}_+^d \mid \|p\| = 1\}$ be the set of probability distributions over $d$ outcomes, represented as vectors. Let relint$(\Delta_d) = \Delta_d \setminus \{p \in \Delta_d | \exists\, i \in [d], p_i = 0\}$. We let $\|\cdot\| : \mathbb{R}^d \to \mathbb{R}_+$ denote a general p-norm. Given a norm $\|\cdot\|$, we define it's dual norm $\|y\|_* = \sup_{\|x\| \leq 1} \langle x, y \rangle$. Let $f : \mathbb{R}^d \to (-\infty, +\infty]$ be a function. We define the Fenchel conjugate of $f$ by $f^* : \mathbb{R}^d \to [-\infty, +\infty]$ such that $f^*(y) = \sup_{x \in \text{dom}(f)} \langle y, x \rangle - f(x)$. We will use the following definitions for a function $f$.

- **convex**: $\forall\, x, y \in \mathbb{R}^d, \lambda \in [0, 1], f(\lambda x + (1 - \lambda)y) \leq \lambda f(x) + (1 - \lambda)f(y)$.

- **increasing**: $f(q) > (q') \,\forall\, q, q' \in \mathbb{R}^d$ with $q \gneq q'$.

- **1-invariant**: $f(q + \alpha \mathbb{1}) = f(q) + \alpha$ for $q \in \mathbb{R}^d, \alpha \in \mathbb{R}$.

- **probability mapping**: $f$ is twice-differentiable, $\nabla f : \mathbb{R}^d \to \Delta_d$, and closure$(\{\nabla f(q) \mid q \in \mathbb{R}^d\}) = \Delta_d$.

We shall refer to a function which is convex, increasing, 1-invariant, and a probability mapping as **CIIP**.

### 2.1 Bregman Divergences, Smoothness, and Strong Convexity

This section presents core concepts used throughout this work. We emphasize that we define both L-smoothness and K-strong convexity through a general norm $\|\cdot\|$, not just via the 2-norm, as is common in some literature.

**Lemma 1** (Hiriart-Urruty and Lemaréchal [2004], Proposition 6.1.1)**.** *For a convex differentiable function $f : \mathbb{R}^d \to (-\infty, +\infty]$, it holds that $\langle \nabla f(x) - \nabla f(y), x - y \rangle \geq 0 \,\forall\, x, y \in \mathbb{R}^d$.*

**Definition 1** (Bregman divergence, L-smoothness, & K-strongly convex)**.** *For a differentiable function $f : \mathbb{R}^d \to (-\infty, +\infty]$, we define the Bregman divergence as $D_f(x, y) = f(x) - [f(y) + \langle \nabla f(y), x - y \rangle]$. If $f : \mathbb{R}^d \to (-\infty, +\infty]$ is convex and differentiable, we say that $f$ is L-smooth w.r.t. $\|\cdot\|$ if $\forall\, x, y \in \mathbb{R}^d$ it holds that $D_f(x, y) \leq \frac{L}{2}\|x - y\|^2$ where $L \geq 0$. If $f : \mathbb{R}^d \to (-\infty, +\infty]$ is convex and differentiable, we say that $f$ is K-strongly convex w.r.t. $\|\cdot\|$ if $\forall\, x, y \in \mathbb{R}^d$ it holds that $\frac{K}{2}\|x - y\|^2 \leq D_f(x, y)$ where $K \geq 0$.*

Conceptually, K-strongly convex acts as a quadratic lower bound of a convex function, while L-smoothness serves as a quadratic upper bound of a convex function. Finally, we present a theorem demonstrating the dual relationship between strong convexity and smoothness.

**Theorem 1** (Shalev-Shwartz [2009]). *Assume that $f$ is a closed and convex function. Then $f$ is K-strongly convex w.r.t. a norm $\|\cdot\|$ i.f.f. $f^*$ is $\frac{1}{K}$-smooth w.r.t. the dual norm $\|\cdot\|_*$.*

## 2.2 Automated Market Makers for AD Securities

We first introduce the general framework for an automated market maker.

**Definition 2** (Automated Market Maker for AD Securities). *Say we have a finite random variable $\mathcal{Y} = \{y_1, \ldots, y_d\}$ over $d$ mutually exclusive and exhaustive outcomes. An automated market maker for AD securities, with initial state $q_0 \in \mathbb{R}^d$, operates as follows. At round $t \in \mathbb{N}_0$,*

*1. A trader can request any bundle of securities $r_t \in \mathbb{R}^d$.*

*2. The trader pays the market maker some amount $Pay(q_t, r_t) \in \mathbb{R}$ in cash.*

*3. The market state updates to $q_{t+1} = q_t + r_t$.*

*After an outcome of the form $Y = y_i$ occurs, for each round $t$, the trader responsible for the trade $r_t$ is paid $(r_t)_i$ in cash, i.e. the number of shares purchased in outcome $y_i$. The market payout for the bundle $r_t$ and the outcome $Y = y$ is expressed via $\langle r_t, \rho(y) \rangle$ where $\rho : \mathcal{Y} \to \delta_y$. At any state $q_t$, the market maker can infer the belief of the market via the instsantaneous price $InstPrice(q_t) = f(q_t, 0)$ where $f(q_t, r_t) = \nabla_{r_t} Pay(q_t, r_t)$.*

Intuitively, the instantaneous price of security $i$ is the amount that would be paid for an arbitrarily small amount of security $i$. We now define a particular family of automated market makers, which captures the DCFMM and the Smooth Quadratic Prediction Market.

**Definition 3** (Price-Plus-Fee Market). *A Price-Plus-Fee Market is an automated market maker of the following form. Let $C : \mathbb{R}^d \to \mathbb{R}$ be **CIIP**. Then*

$$Pay(q_t, r_t) = \langle \nabla C(q_t), r_t \rangle + Fee(q_t, r_t) = \langle p_t, r_t \rangle + Fee(q_t, r_t)$$

*where $Fee(q_t, r_t) = o(\|r_t\|)$. We note that in this case, $InstPrice(q_t) = p_t = \nabla C(q_t)$.*

Given that both markets we examine in this work are Price-Plus-Fee markets, we shall use $\nabla C(q_t)$ and $InstPrice(q_t)$ interchangeably.

Generally, it is common practice to expect prediction markets to satisfy some form of the following axioms.

**Axiom 1.** *(Existence of Instantaneous Price): $C$ is continuous and differentiable everywhere on $\mathbb{R}^d$.*

**Axiom 2.** *(Information Incorporation): for any $q, r \in \mathbb{R}^d$, $Pay(q + r, r) \geq Pay(q, r)$.*

**Axiom 3.** *(No Arbitrage) For a Price-Plus Fee market, no arbitrage is defined such that for any given trade sequence $s = r_0, \ldots r_i \in (\mathbb{R}^d)^*$ and initial state $q_0 \in \mathbb{R}^d$ it holds that*

$$\min_{y \in \mathcal{Y}} \langle \rho(y), q_0 + \sum_{i=0}^{i-1} r_i \rangle \leq \sum_{i=0}^{i-1} Pay(q_i, r_i) .$$

*For the case where $s = \emptyset$, equality must hold in the previous inequality.*

**Axiom 4.** *(Expressiveness): For any $p \in \Delta_d$ and $\epsilon > 0$, $\exists q \in \mathbb{R}^d$ such that $\|InstPrice(q) - p\| < \epsilon$.*

**Axiom 5.** *(Incentive Compatibility): Assume that the market is at state $q_t$ and that the agent has a belief $\mu \in \Delta_d$. To maximize expected return*

$$\arg\max_{r_t \in \mathbb{R}^d} \underbrace{\langle \mu, r_t \rangle}_{\text{Expected Payout}} - \underbrace{Pay(q_t, r_t)}_{\text{Payment to Market}} , \tag{1}$$

*the agent will purchase a bundle $r_t$ such that for $q_{t+1} = q_t + r_t$ it holds that $InstPrice(q_{t+1}) = \mu$.*

**Intuition of Axioms** Axiom 1 states that any market state can mapped to a distribution. Axiom 2 states that if a trader were to purchase the same bundle twice, the price would be larger the second time. Axiom 3 states that for any bundle purchase there exists an outcome where the trader losses money. Axiom 4 states that the state of the market can mapped to a distribution that expresses a belief arbitrarily close. Axiom 5 states that a trader desires to move the market to a state such that $\nabla C$ maps the market state to their belief of the distribution of $\mathcal{Y}$. See Abernethy et al. [2013] for further discussion regarding the intuition of axioms.

## 2.3 CIIP Construction

It may seem unclear how one could construct a $C$ which is **CIIP**. We later show that via a $C$ being **CIIP**, a Price-Plus-Fee Market is able to automatically satisfy many of our desired axioms In this subsection we provide supporting lemmas for constructing **CIIP** functions and provide some examples.

**Lemma 2** (Abernethy et al. [2013], Theorem 4.2, Lemma 4.3). *Let $\hat{C} : \mathbb{R}^d \to \mathbb{R}$ be defined over relint$(\Delta_d)$, $\hat{C}$ be strictly convex over its domain, and define $C = \hat{C}^*$. As $\hat{C}$ is strictly convex, $\nabla C(q) = \arg\max_{p \in dom(\hat{C})} \langle q, p \rangle - \hat{C}(p)$. Furthermore, assuming $dom(\hat{C})$ is restricted to either relint$(\Delta_d)$ or $\Delta_d$, we also have closure$(\{\nabla C(q) \mid q \in \mathbb{R}^d\}) = \Delta_d$.*

Note, if $\hat{C} : \mathbb{R}^d \to \mathbb{R}$ was $\frac{1}{L}$-strongly convex w.r.t. $\|\cdot\|$ then $C$ would be $L$-smooth w.r.t. $\|\cdot\|_*$ via Theorem 1.

A $C$ which is one-invariant allows for $\text{InstPrice}(q_t) = \text{InstPrice}(q_{t+1})$ where $q_{t+1} = q_t + \alpha\mathbb{1}$ such that $\alpha \in \mathbb{R}$. This is a desirable property since the purchase of $q_{t+1} = q_t + \alpha\mathbb{1}$ incorporates information uniformly.

**Lemma 3.** *W.r.t. a function $f$ restricted to the $\Delta_d$ (or relint$(\Delta_d)$), the Fenchel conjugate $f^*$ is one-invariant.*

*Proof.* Let $q \in \mathbb{R}^d$ and $\alpha \in \mathbb{R}$. Observe the following $f^*(q + \alpha\mathbb{1}) = \sup_{p \in \Delta_d} \langle q + \alpha\mathbb{1}, p \rangle - f(p) = \sup_{p \in \Delta_d} \langle q, p \rangle - f(p) + \alpha\langle\mathbb{1}, p\rangle = f^*(q) + \alpha$. $\square$

We proceed to give examples of $\hat{C}$ and the corresponding $C$ which satisfies **CIIP**. For the case of $\hat{C}(p) = L^{-1} \sum_{i=1}^d p_i \log p_i$, the derived $C$ is softmax.

**Softmax** Let $\hat{C}(p) = L^{-1} \sum_{i=1}^d p_i \log p_i$. Then $C(q) = \sup_{p \in \text{relint}(\Delta_d)} \langle q, p \rangle - \hat{C}(p) = L^{-1} \log(\sum_{i=1}^n e^{Lq_i})$. Furthermore, $\nabla C(q) = \arg\max_{p \in \text{relint}(\Delta_d)} \langle q, p \rangle - \hat{C}(p) = \frac{e^{Lq}}{\sum_{i=1}^n e^{Lq_i}}$ where $C(q)$ is L-smooth w.r.t. $\|\cdot\|_2$ and $\|\cdot\|_\infty$. Also, it holds that $\{\nabla C(q) \mid q \in \mathbb{R}^d\} = \text{int}(\Delta_d)$ and thus closure$(\{\nabla C(q) \mid q \in \mathbb{R}^d\}) = \Delta_d$.

For the case of $\hat{C}(p) = \frac{L}{2}\|p\|_2^2$, the derived $C$ is referred to in the machine learning literature as sparsemax [Martins and Astudillo, 2016, Niculae and Blondel, 2017].

**Sparsemax** Let $\hat{C}(p) = \frac{L}{2}\|p\|_2^2$ for $L > 0$. Then $C(q) = \sup_{p \in \Delta_d} \langle q, p \rangle - \hat{C}(p) = \sup_{p \in \Delta_d} \langle q, p \rangle - \frac{L}{2}\|p\|_2^2$. Furthermore, $\nabla C(q) = \arg\max_{p \in \Delta_d} \langle q, p \rangle - \hat{C}(p) = \arg\min_{p \in \Delta_d} \|q - \frac{p}{L}\|_2^2$ where $C(q)$ is $\frac{1}{L}$-smooth w.r.t. $\|\cdot\|_2$. Also, it holds that $\{\nabla C(q) \mid q \in \mathbb{R}^d\} = \Delta_d$ and thus closure$(\{\nabla C(q) \mid q \in \mathbb{R}^d\}) = \Delta_d$.

## 2.4 Duality-based CFMM

We now define the DCFMM which provides a construction scheme for a cost function $C$ respectively and a particular $\text{Pay}(\cdot, \cdot)$ scheme.

**Definition 4** (Abernethy et al. [2013], DCFMM for AD Securities). *Let $C : \mathbb{R}^d \to \mathbb{R}$ be **CIIP** such that $C := \hat{C}^*$ where $\hat{C}$ is strictly convex and continuous over all of relint$(\Delta_d)$. At state $q_t$ for bundle $r_t$, we define the payment of a DCFMM by $\text{Pay}_D(q_t, r_t) = C(q_t + r_t) - C(q_t)$.*

With respect to the Price-Plus-Fee Market (Definition 3), one could think of the DCFMM market payment as a linear term plus an implicit fee based on the Bregman divergence by

$$\text{Pay}_D(q_t, r_t) = C(q_{t+1}) - C(q_t) = \langle p_t, r_t \rangle + \underbrace{D_C(q_{t+1}, q_t)}_{\text{Breg.-Fee}}$$

since $D_C(q_{t+1}, q_t) = C(q_{t+1}) - C(q_t) - \langle p_t, r_t \rangle$ and $q_{t+1} = q_t + r_t$. Intuitively, the DCFMM charges the instantaneous price per share in a bundle $r_t$ via $\langle p_t, r_t \rangle$ and then charges a a "curvature fee" via the Bregman Fee to hedge the potential loss for large trades. The DCFMM satisfies Axioms 1-5, which we now show.

**Theorem 2** (Abernethy et al. [2013], Theorem 3.2). *The DCFMM satisfies Axioms 1-4.*

Trivially, with $\text{Pay}_D$ plugged into Eq (1) and taking the gradient with respect to $q_{t+1}$ and setting it equal to zero, we can see that Axiom 5 is also satisfied for the DCFMM. Furthermore, the DCFMM has the nice property that the worst-case loss of the market is bounded.

**Theorem 3** (Abernethy et al. [2013], Theorem 4.4). *The DCFMM has a worst-case loss no more than* $\sup_{p \in \rho(\mathcal{Y})} \hat{C}(p) - \min_{p \in \Delta_d} \hat{C}(p)$.

As noted by Abernethy et al. [2013], FTRL's regrate rate and DCFMM's worst-case loss are similar. We show in Appendix C the equivalence to Continuous FTRL's regret rate and DCFMM's worst-case loss.

## 3 Smooth Quadratic Prediction Markets

### 3.1 Smooth Quadratic Prediction Market Design

We now introduce the Smooth Quadratic Prediction Market. Given a smooth **CIIP** function $C$ w.r.t. a general norm $\|\cdot\|$, we propose charging a fee based on the upper quadratic bound obtained via the $L$-smoothness condition.

**Definition 5** (Smooth Quadratic Prediction Market). *Let* $C : \mathbb{R}^d \to \mathbb{R}$ *be* **CIIP**. *Assume that* $C$ *is L-smooth w.r.t.* $\|\cdot\|$. *At state* $q_t$ *for bundle* $r_t$, *we define the payment of a Smooth Quadratic Prediction Market by*

$$Pay_L(q_t, r_t) = \langle p_t, r_t \rangle + \underbrace{\frac{L}{2}\|r_t\|^2}_{Q\text{-}Fee} .$$

In the remainder of this section, we show that the Smooth Quadratic Prediction Market satisfies Axioms 1-4 and has a better worst-case loss than the DCFMM. Note, we don't claim that the Smooth Quadratic Prediction Market satisfies Axiom 5 Incentive Compatibility. Later in Section 3.2 and 3.3, we show that the traders within the Smooth Quadratic Prediction Market satisfy instead a form of *incremental* incentive compatibility, and interestingly, the traders mimic the update steps of gradient (general) steepest descent while trading.

**Lemma 4.** *By the assumption that* $C$ *is* **CIIP**, *the Smooth Quadratic Prediction Market satisfies Axiom 1 Existence of Instantaneous Price.*

**Lemma 5.** *The Smooth Quadratic Prediction Market satisfies Axiom 2 Information Incorporation.*

*Proof.* Since $C$ is convex and differentiable, by the monotonicity of gradients (Lemma 1), we have that

$$\langle \nabla C(q_{t+1}) - \nabla C(q_t), q_{t+1} - q_t \rangle \geq 0 \Leftrightarrow \langle \nabla C(q_t + r_t), r_t \rangle \geq \langle \nabla C(q_t), r_t \rangle$$

where $q_{t+1} = q_t + r_t$. Then by adding $\frac{L}{2}\|r_t\|^2$ to both sides we get

$$\langle \nabla C(q_t + r_t), r_t \rangle + \frac{L}{2}\|r_t\|^2 \geq \langle \nabla C(q_t), r_t \rangle + \frac{L}{2}\|r_t\|^2 \Leftrightarrow \text{Pay}_L(q_t + r_t, r_t) \geq \text{Pay}_L(q_t, r_t) .$$

$\square$

**Lemma 6.** *The Smooth Quadratic Prediction Market satisfies Axiom 3 No Arbitrage.*

*Proof.* By the definition of $\text{Pay}_D$ and $\text{Pay}_L$, we have that for any sequence $s = q_0, \ldots, q_i$ (including the empty sequence as both pay functions are equal to zero) that

$$\sum_{i=0}^{i-1} \text{Pay}_D(q_i, r_i) \leq \sum_{i=0}^{i-1} \text{Pay}_L(q_i, r_i) .$$

By (Theorem 3.2, Abernethy et al. [2013]) the DCFMM satisfies no arbitrage, we have that

$$\min_{y \in \mathcal{Y}} \langle \rho(y), q_0 + \sum_{i=0}^{i-1} r_i \rangle \leq \sum_{i=0}^{i-1} \text{Pay}_D(q_i; r_i) .$$

Therefore by combing the two inequalities we have that the claim holds. $\qquad\square$

**Lemma 7.** *The Smooth Quadratic Prediction Market satisfies Axiom 4 Expressiveness.*

*Proof.* By the **CIIP** assumption on $C$, it holds that closure($\{\nabla C(q) \mid q \in \mathbb{R}^d\}) = \Delta_d$ which states every limit point of the set $\{\nabla C(q) \mid q \in \mathbb{R}^d\}$ is in $\Delta_d$. Hence, by definition of Expressiveness, the claim holds. $\qquad\square$

**Theorem 4.** *For any fixed trade history $h = (r_0, \ldots, r_t)$, the Smooth Quadratic Prediction Market has a better worst-case loss then the DCFMM.*

*Proof.* Observe for any $(q_i, r_i)$ for $i \in 0 \ldots, t$ that

$$C(q_i + r_i) - C(q_i) = \langle p_i, r_i \rangle + D_C(q_{i+1}, q_i) \leq \langle p_i, r_i \rangle + \frac{L}{2} \|r_i\|^2$$

via Definition 1. Hence, the collected revenue of the the Smooth Quadratic Prediction Market is greater overall than that of the DCFMM implying a better worst-case loss then the DCFMM. $\qquad\square$

### 3.2 $\ell_2$-based Smooth Quadratic Prediction Markets

As mentioned, the core purpose of a prediction market with AD securities is to elicit a distribution over a finite set of outcomes. Hence, it is of essence to show that traders are incentivized to move the market state such that $\nabla C$ maps the market state to a trader's belief; however, recall that we stated the Smooth Quadratic Prediction Market does not satisfy the Axiom of Incentive Compatibility. Interestingly, the expectation maximizing trading behavior of an agent w.r.t. the $\ell_2$-based Smooth Quadratic Prediction Market is expressible via gradient descent. We show that the $\ell_2$-based Smooth Quadratic Prediction Market satisfies a form of *incremental* incentive compatibility. Overall, the core result of this section, Theorem 6, states that a trader is incentivized to move the market state such that the market state maps to their belief via a sequence of bundle purchases instead of via a single transaction. We formally define incremental incentive compatibility as follows.

**Axiom 6.** *(Incremental Incentive Compatibility): Assume the market is at state $q_0$ and that a sequence of agents with the same belief $\mu \in \Delta_d$ purchases bundles $r_t$ relative to maximizing their expected payout*

$$\arg\max_{r_t \in \mathbb{R}^d} \quad \underbrace{\langle \mu, r_t \rangle}_{\textit{Expected Payout}} \quad - \quad \underbrace{\textit{Pay}(q_t, r_t)}_{\textit{Payment to Market}} .$$

*Then $\lim_{t \to \infty} \nabla C(q_t) = \mu$.*

We now formally define gradient descent (GD) and provide a supporting Theorem and Lemma for GD which will be used in proving incremental incentive compatibility for the $\ell_2$-based Smooth Quadratic Prediction Market.

**Definition 6** (Gradient Descent). *Let $x_0 \in \mathbb{R}^d$, and let $\gamma > 0$ be a step size. Given a differentiable function $f$, the gradient descent (GD) algorithm defines a sequences $(x_t)_{t \in \mathbb{N}_0}$ satisfying $x_{t+1} = x_t - \gamma \nabla f(x_t)$.*

**Theorem 5** (Garrigos and Gower [2023], Theorem 3.4). *Assume that $f$ is convex and $\ell_2$-based $L$-smooth, for some $L > 0$. Let $(x_t)_{t \in \mathbb{N}_0}$ be the sequence of iterates generated by the GD algorithm, with a stepsize satisfying $0 < \gamma \leq \frac{1}{L}$. Then, for all $x^* \in \arg\min f$ and for all $t \in \mathbb{N}_0$ we have that*

$$f(x^{(t)}) - \inf f \leq \frac{\|x^{(0)} - x^*\|_2^2}{2\gamma t} .$$

**Lemma 8** (Sidford [2024], Lemma 6.1.6). *If $f : \mathbb{R}^d \to \mathbb{R}$ is $L$-smooth w.r.t. $\|\cdot\|$ then for all $x^* \in \arg\min f$ and $x \in \mathbb{R}^d$ it holds that*

$$\frac{1}{2L}\|\nabla f(x)\|_*^2 \le f(x) - f(x^*) \le \frac{L}{2}\|x - x^*\|^2 \, .$$

**Theorem 6.** *With respect to some **CIIP** $C$, define a $\ell_2$-based Smooth Quadratic Prediction Market. The market satisfies Axiom 6 Incremental Incentive Compatibility furthermore $\lim_{t\to\infty} \nabla C(q_t) = \mu$ at a rate of $\frac{1}{t}$.*

*Proof.* Let $\overline{C}(q) = C(q) - \langle \mu, q \rangle$ and note that $\nabla \overline{C}(q) = \nabla C(q) - \mu$. Note that the utility of an agent is equivalent to the following

$$\min_{q_{t+1} \in \mathbb{R}^d} \quad \underbrace{\left( \langle \nabla C(q_t), q_{t+1} - q_t \rangle + \frac{L}{2}\|q_{t+1} - q_t\|_2^2 \right)}_{\text{Payment to Market}} - \underbrace{\langle \mu, q_{t+1} - q_t \rangle}_{\text{Expected Payout}}.$$

Hence, the update of the market state $q_{t+1} \leftarrow q_t - (\frac{1}{L})(\nabla C(q_t) - \mu) = q_t - (\frac{1}{L})\nabla \overline{C}(q_t)$ is a GD step performed on $\overline{C}$. Observe, setting $\nabla \overline{C}(q) = 0$ we get that $\nabla C(q) = \mu$, i.e. for $q^* \in \arg\min \overline{C}(q)$ it holds that $\nabla C(q^*) = \mu$. Hence, $q_{t+1} \leftarrow q_t - (\frac{1}{L})(\nabla C(q_t) - \mu) = q_t - (\frac{1}{L})\nabla \overline{C}(q_t)$ is a GD update for $\overline{C}(q) = C(q) - \langle \mu, q \rangle$ with learning rate $\frac{1}{L}$ whose minimum point is a $q^*$ such that $\nabla C(q^*) = \mu$. Also note that if $\nabla C(q_t) = \mu$ then $q_{t+1} \leftarrow q_t - (\frac{1}{L})(\nabla C(q_t) - \mu) = q_t - (\frac{1}{L})\nabla \overline{C}(q_t)$ hence, any minimizer of $\overline{C}$ is a stationary point. Observe that $\overline{C}$ is $\ell_2$-based smooth and hence Theorem 5 and Lemma 8 apply. $\qquad\square$

### 3.3 $\ell_p$-based Smooth Quadratic Prediction Markets

We now generalize our results for the Smooth Quadratic Prediction Market defined w.r.t. $\|\cdot\|_p$ for $p \in [1, \infty]$. By altering the norm, the Smooth Quadratic Prediction Market gets varying behavior pertaining to the charge for Q-fee and path of convergence of $\{q_t\}_{t\in\mathbb{N}_0} \to \arg\min_{q\in\mathbb{R}^d} \overline{C}(q)$. In this generalized setting, agents are incentivized to trade following the general steepest descent algorithm in order to maximize their expected return. We now formally define general steepest descent (SD) and provide a supporting Theorem for SD which will be used in proving incremental incentive compatibility for the $\ell_p$-based Smooth Quadratic Prediction Market.

**Definition 7** (General Steepest Descent). *Given a function $f$ that is differentiable and $L$-smooth w.r.t. $\|\cdot\|$, the general steepest descent (SD) method w.r.t. the norm $\|\cdot\|$ iteratively defines a sequences $(x_t)_{t\in\mathbb{N}_0}$ via $x_{t+1} \in \arg\min_{x\in\mathbb{R}^d} \langle \nabla f(x_t), x - x_t \rangle + \frac{L}{2}\|x - x_t\|^2$.*

**Theorem 7** (Kelner et al. [2014], Theorem 1). *If $f$ is convex and $L$-smooth w.r.t. $\|\cdot\|$, then SD w.r.t. $\|\cdot\|$ satisfies $f(x_t) - \inf f \le \frac{2LK^2}{t+4}$ with $K := \max_{x:f(x)\le f(x_0)} \min_{x^*:f(x^*)=\inf f} \|x - x^*\|$ where $(x_t)_{t\in\mathbb{N}_0}$ is a sequence of iterates generated by the SD algorithm.*

**Theorem 8.** *With respect to some **CIIP** $C$, define an $\ell_p$-based Smooth Quadratic Prediction Market. The market satisfies Axiom 6 Incremental Incentive Compatibility furthermore $\lim_{t\to\infty} \nabla C(q_t) = \mu$ at a rate of $\frac{1}{t}$.*

*Proof.* Observe that the utility of an agent is equivalent to the following

$$\begin{aligned}
&\arg\min_{q_{t+1} \in \mathbb{R}^d} \quad \underbrace{\left( \langle \nabla C(q_t), q_{t+1} - q_t \rangle + \frac{L}{2}\|q_{t+1} - q_t\|_p^2 \right)}_{\text{Payment to Market}} - \underbrace{\langle \mu, q_{t+1} - q_t \rangle}_{\text{Expected Payout}} \\
&\Leftrightarrow \arg\min_{q_{t+1} \in \mathbb{R}^d} \quad \langle \nabla \overline{C}(q_t), q_{t+1} - q_t \rangle + \frac{L}{2}\|q_{t+1} - q_t\|_p^2
\end{aligned} \tag{2}$$

as $q_{t+1} = q_t + r_t$. Hence, Eq. (2) is a SD step for $\overline{C} = C(q) - \langle \mu, q \rangle$. Observe that $\overline{C}$ is $\ell_p$-based smooth and hence Theorem 7 and Lemma 8 apply. $\qquad\square$

In Figure 1, we demonstrate the varying convergence behavior of the Smooth Quadratic Prediction Market when using different norms. There are numerous convergence results for gradient (general

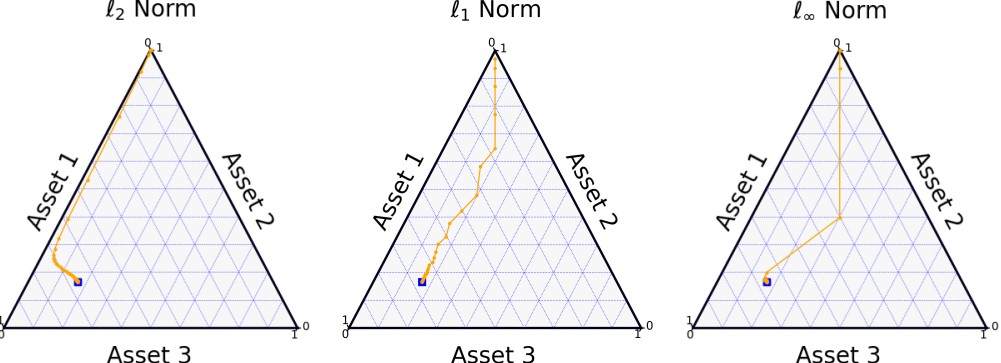

Figure 1: Let $q_0 = (10, 20, 10)$, $C$ is softmax with smoothness of $L = 1$, and $\mu = (1/6, 1/6, 2/3)$. The blue square expresses $\mu$ and the orange path towards the blue square demonstrates the updating market distribution states. As denote by the titles's of each plot, we vary the norm used for the Smooth Quadratic Prediction Market. Note although softmax is not $\ell_1$-smooth, we use said norm experimentally for the sake of comparison.

steepest) descent. To avoid redundancy with the literature, we refer the reader to Appendix D for further convergence results, which may be of interest to the application of Smooth Quadratic Prediction Markets.

## 4 Smooth Quadratic Prediction Markets with Constraints

In this section, we examine the trading behavior of expectation-maximizing agents under two realistic constraints (independently): budget-bounded traders and buy-only markets. For mathematical sake, we solely examine the $\ell_2$-based Smooth Quadratic Prediction Market buy-only market and perform experiments for the budget-bounded trader setting. Furthermore, for both types of constraints, we perform simulations for the $\ell_1$ and $\ell_\infty$ cases. Overall, either analytically or experimentally, we observe that in both situations, the market state converges to the belief of a sequence of traders, regardless of the used p-norm, motivating the use of the Smooth Quadratic Prediction Market in either of these realistic scenarios.[*] We refer the reader to Appendix E for the omitted analysis within this section.

**Budget-Bounded Traders**  Following the work of Fortnow and Sami [2012], we consider the notation of *natural budget constraint*, which states that the loss of the agent is at most their budget, for all $y \in \mathcal{Y}$. The objective of an agent is expressed by

$$\min_{r_t \in \mathbb{R}^d} \quad \underbrace{\left( \langle \nabla C(q_t), r_t \rangle + \frac{L}{2} \|r_t\|_2^2 \right)}_{\text{Payment to Market}} - \underbrace{\langle \mu, r_t \rangle}_{\text{Expected Payout}}$$

$$\text{s.t.} \quad \underbrace{\left( \langle \nabla C(q_t), r_t \rangle + \frac{L}{2} \|r_t\|_2^2 \right)}_{\text{Payment to Market}} - \underbrace{\langle \delta_y, r_t \rangle}_{\text{Realized Payout}} \leq \underbrace{B}_{\text{Budget}}, \quad \forall\, y \in \mathcal{Y}$$

where $B \in \mathbb{R}_{>0}$. Experimentally, using $\nabla C$ equal to softmax, we were able to observe convergence regardless of the initial state, belief distribution, and budget used. The rate of convergence was proportional to the size of the budget. Figure 2 in Appendix E demonstrates the convergence behavior of the market under budget constraints.

**Buy-Only Market**  The Buy-Only Market Li and Vaughan [2013] assumes that $0 \preceq r_t$ for all $t \in \mathbb{N}_0$; hence, only positive bundles purchases are allowed. The objective of the trader is expressed

---

[*]Code can be found at: https://github.com/EnriqueNueve/Smooth-Quadratic-Prediction-Markets

via the equation below.

$$\min_{r_t \in \mathbb{R}^d} \quad \underbrace{\left( \langle \nabla C(q_t), r_t \rangle + \frac{L}{2} \|r_t\|_2^2 \right)}_{\text{Payment to Market}} - \underbrace{\langle \mu, r_t \rangle}_{\text{Expected Payout}}$$

$$\text{s.t.} \quad \underbrace{0 \preceq r_t}_{\text{Buy Only Constraint}}$$

From solving the Lagrange dual problem, we get that the update step is $q_{t+1} = q_t + \frac{1}{L}((\nabla C(q_t) - \mu)_+ - (\nabla C(q_t) - \mu))$ and observe when $\nabla C(q_t) = \mu$ we have a stationary point. Observe the update is a coordinate GD update for $\overline{C}$ when $(\nabla C(q_t) - \mu)_i < 0$ and $q_{t+1,i} = q_{t,i}$ when $(\nabla C(q_t) - \mu)_i \geq 0$. Hence, not just experimentally but analytically, we are able to show incremental incentive compatibility.

# 5 Smooth Quadratic Prediction Markets with Adaptive Liquidity

Inspired by the paper *A General Volume-Parameterized Market Making Framework* [Abernethy et al., 2014], we adapt aspects of the framework to our setting to facilitate adaptive liquidity. Liquidity in this context can be thought of as the sensitivity of price changes with respect to bundle purchases. A high liquidity would indicate that a large bundle purchase would not cause a significant price change and vice-versa. Adaptive liquidity can be facilitated by decreasing the smoothness of $C$ as the volume of trades increases.

Let $\mathbf{S} = (\mathbb{R}^d)^*$ denote the history space of the market consisting of finite (and possibly empty) sequences of bundles. We define a volume update function via an asymmetric norm.

**Definition 8** (VPM). *Say we have a finite random variable $\mathcal{Y} = \{y_1, \ldots, y_d\}$ over $d$ mutually exclusive and exhaustive outcomes. Let $C^\circ : \mathbb{R}^d \times \mathbb{R}_+ \to \mathbb{R}$ be **CIIP** and increasing in $v$ (it's second argument). Assume $C^\circ$ is L-smooth w.r.t. some general norm $\|\cdot\|$. Furthermore, as $v$ increases, the smoothness of $C^\circ$ should decrease. Let $g : \mathbb{R}^d \to \mathbb{R}_+$ be an asymmetric norm (refer to Def. 15 in Appendix F) and define w.r.t. a $s = r_0 \ldots, r_t \in \mathbf{S}$ the volume update function $V(s) = v_0 + \sum_{i=0}^t g(r_i)$. The VPM for AD securities defined by $C^\circ$, with initial state $q_0$, and initial volume $v_0 \in \mathbb{R}_+$ operates as follows. At round $t \in \mathbb{N}_0$,*

*1. A trader can request any bundle of securities $r_t \in \mathbb{R}^d$.*

*2. The trader pays the market maker some amount $Pay(q_t, r_t; v_t) \in \mathbb{R}$ (such that Pay is dependent on $C^\circ$) in cash.*

*3. The market state updates to $q_{t+1} = q_t + r_t$ and $v_{t+1} = V(s)$ such that $s = r_0, \ldots, r_t$.*

*After an outcome of the form $Y = y_i$ occurs, for each round $t$, the trader responsible for the trade $r_t$ is paid $(r_t)_i$ in cash, i.e. the number of shares purchased in outcome $y_i$. The market payout for the bundle $r_t$ and the outcome $Y = y$ is expressed via $\langle r_t, \rho(y) \rangle$ where $\rho : \mathcal{Y} \to \delta_y$. At any state $(q_t, v_t)$, the market maker can infer the belief of the market via the instsantaneous price $InstPrice(q_t; v_t) = f(q_t, 0; v_t)$ where $f(q_t, r_t; v_t) = \nabla_{r_t} Pay(q_t, r_t; v_t)$.*

For the case of the VPM-DCFMM, the pay function is the following

$$\text{Pay}_{D^\circ}(q_t, r_t; v_t) = C^\circ(q_t + r_t; v_t + g(r_t)) - C^\circ(q_t; v_t).$$

The following inequality due to the smoothness of $C^\circ$ motivates our proposed payment of $\text{Pay}_{L^\circ}$

$$
\begin{aligned}
\text{Pay}_{D^\circ}(q_t, r_t; v_t) &= C^\circ(q_t + r_t; v_t + g(r_t)) - C^\circ(q_t; v_t) \\
&= \underbrace{\Big( C^\circ(q_t + r_t; v_t + g(r_t)) - C^\circ(q_t; v_t + g(r_t)) \Big)}_{\text{Payment to Breg. Market}} - \underbrace{\Big( C^\circ(q_t; v_t + g(r_t)) - C^\circ(q_t; v_t) \Big)}_{\text{Liquidity Fee} \geq 0} \\
&= \underbrace{\Big( \langle \nabla C^\circ(q_t; v_t + g(r_t)) \rangle + D_{C^\circ(\cdot; v_t + g(r_t))}(q_t + r_t, q_t) \Big)}_{\text{Payment to Breg. Market}} - \underbrace{\Big( C^\circ(q_t; v_t + g(r_t)) - C^\circ(q_t; v_t) \Big)}_{\text{Liquidity Fee} \geq 0} \\
&\leq \underbrace{\Big( \langle \nabla C^\circ(q_t; v_t + g(r_t)), r_t \rangle + \frac{L^\circ}{2} \| r_t \|^2 \Big)}_{\text{Payment to Smooth-Quad Market}} + \underbrace{\Big( C^\circ(q_t; v_t + g(r_t)) - C^\circ(q_t; v_t) \Big)}_{\text{Liquidity Fee} \geq 0} \\
&= \text{Pay}_{L^\circ}(q_t, r_t; v_t).
\end{aligned}
$$

Using this approach, we can easily show that $\text{Pay}_{L^\circ}(q_t, r_t; v_t)$ facilitates no arbitrage (proof in Appendix F).

**Lemma 9.** *$Pay_{L^\circ}$ satisfies Axiom 3 No Arbitrage.*

However, proving other important properties such as bounded-worst case loss, information incorporation, and some form of incentive compatibility becomes challenging due to the non-convexity of the expected return

$$
\arg\max_{r_t \in \mathbb{R}^d} \quad \underbrace{\langle \mu, r_t \rangle}_{\text{Expected Payout}} - \underbrace{\text{Pay}_{L^\circ}(q_t, r_t; v_t)}_{\text{Payment to Market}}.
$$

We leave proving said results to future work. So, although this approach facilitates adaptive liquidity and no arbitrage, more work is required to justify its use.

# 6 Conclusion

**Recap** In this work, we proposed a new prediction market framework based on the traditional DCFMM framework. By doing so, the Smooth Quadratic Prediction Market satisfies many of the same axioms of the DCFMM while facilitating higher profits for the market maker. Although the Smooth Quadratic Prediction does not satisfy the axiom of incentive compatibility, we show that the axiom is satisfied in an incremental sense and interestingly relate the methodology of general steepest descent with said behavior. We also examined the Smooth Quadratic Prediction Market under the constraints of budget-bounded traders and buy-only markets. Finally, we also presented introductory work on how the Smooth Quadratic Prediction Market could facilitate adaptive liquidity. Although this work provides some core insights into the properties of the Smooth Quadratic Prediction Market, there are many future directions for this work.

**Future Directions** One direction would be to generalize this work beyond AD securities. The original work of Abernethy et al. [2013] demonstrates how the DCFMM can be used for combinatorial and infinite space outcome securities. Given the close relation between the design of the Smooth Quadratic Prediction Market and the DCFMM, we believe this generalization to more securities would be a feasible future direction. Another interesting future direction would be to analyze the convergence of the market when agents have varying beliefs. Using a random selection mechanism to select whose trades get processed, the analysis could be reduced to stochastic gradient descent. Another direction would be to further prove/disprove axiom guarantees regarding our proposed approach to adaptive liquidity. Finally, inspired by the shown equivalence of DCFMM and constant-function market makers (CFMMs) for asset exchanges by Frongillo et al. [2023], one application would be to use the Smooth Quadratic Prediction Market to run an asset exchange.

**Broader Impacts:** Our work informs the design of prediction markets, and thus, our work may influence the choices of prediction market makers and traders. Due to the inherent risk of losing money in prediction markets, we acknowledge the possibility of negative impacts due to this work.

## Acknowledgments and Disclosure of Funding

We thank the following people for their feedback throughout this project: Stephen Becker for talks on gradient descent, Maneesha Papireddygari and Rafael Frongillo for talks about liquidity provisioning, Drona Khurana for talks about convergence results, and Aaron Sidford for notes on general steepest descent.

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

# A  Gradient Theorem of Line Integrals

**Definition 9** (Vector field). *A vector field is a map $F : \mathbb{R}^d \to \mathbb{R}^d$.*

**Definition 10** (Conservative Vector Field). *If the vector field $F$ is the gradient of a function $f$, then $F$ is called a gradient or a conservative vector field. The function $f$ is called the potential or scalar of $F$.*

**Definition 11** (Differentiable Multivariable Function). *The function $f : \mathbb{R}^n \to \mathbb{R}^m$ is differentiable at the point $a$ if there exists a linear transformation $T : \mathbb{R}^n \to \mathbb{R}^m$ that satisfies the condition*

$$\lim_{x \to a} \frac{||f(x) - f(a) - T(x - a)||}{||x - a||} = 0 .$$

*A function $f$ is said to be differentiable if $f$ is differentiable at all points within $dom(f)$.*

**Definition 12** (Line integral of a Vector Field). *For a vector field $F : U \subseteq \mathbb{R}^d \to \mathbb{R}^d$, the line integral along a piecewise smooth curve $s \subset U$, in the direction of $r$, is defined as*

$$\int_s F(r) \cdot dr = \int_a^b \langle F(r(t)), r'(t) \rangle dt$$

*where $\cdot$ is the dot product and $r[a, b] \to s$ is a bijective parametrization of the curve $s$ such that $r(a)$ and $r(b)$ give the endpoints of $s$.*

**Theorem 9** (Gradient Theorem of Line Integrals). *For $\phi : U \subseteq \mathbb{R}^d \to \mathbb{R}$ as a differentiable function and $s$ as any continuous curve in $U$ which starts at a point $p$ and ends at a point $q$, then*

$$\int_s \nabla \phi(r) \cdot dr = \phi(q) - \phi(p)$$

*where $\nabla \phi$ denotes the gradient vector field of $\phi$.*

# B  Continuous FTRL

A variety of works have examined FTRL in the continuous setting (CFTRL) such as Kwon and Mertikopoulos [2017], Mertikopoulos et al. [2018], Cheung and Piliouras [2021]. Let $\mathcal{C} \subseteq \mathbb{R}^d$ denote a non-empty compact convex set.

**Definition 13** (Regularizer). *A convex function $R : \mathbb{R}^d \to \mathbb{R} \cup \{+\infty\}$ will be called a regularizer function on $\mathcal{C}$ if $dom(R) = \mathcal{C}$ and $R|_{\mathcal{C}}$ is strictly convex and continuous.*

For a given regularizer function $R$ on $\mathcal{C}$, we let $R_{max} = \max_{x \in \mathcal{C}} R(x)$ and $R_{min} = \min_{x \in \mathcal{C}} R(x)$.

**Definition 14** (Choice Map). *The choice map associated to a regularizer function $R$ on $\mathcal{C}$ will be the map $Q_R : \mathbb{R}^d \to \mathcal{C}$ defined as*

$$Q_R(y) = \arg\max_{x \in \mathcal{C}} \{\langle x, y \rangle - R(x)\}, \quad y \in \mathbb{R}^d .$$

*Note: $Q_R$ is the convex conjugate of $R$ and we have argmax since $R$ is strict convex and continuous hence has a well-defined unique sup.*

In continuous time, instead of a sequence of payoff vectors $(u_n)_{n \in \mathbb{N}_0}$ in $\mathbb{R}^d$, the agent will be facing a measurable and locally integrable stream of payoff vectors $(u_t)_{t \in \mathbb{R}_+}$ in $\mathbb{R}^d$. Consider the process:

$$x_t^c := Q_R(\eta_t \int_0^t u_s ds) = \arg\max_{x \in \mathcal{C}} \{\langle x, \eta_t \int_0^t u_s ds \rangle - R(x)\},$$

where $(\eta_t)_{t \in \mathbb{R}_+}$ is a positive, nonincreasing and piecewise continuous parameter, while $x_t^c \in \mathcal{C}$ denotes the agent's action at time $t$ given the history of payoff vectors $u_s, 0 \le s < t$. The dynamics of CFTRL can be expressed by

$$\begin{cases} y(t) = y(0) + \int_0^t u(x(s))ds \\ x(t) = Q_R(y_t) \end{cases} .$$

It is worth noting that when $R$ is the entropy function, the dynamics reduce to replicator dynamics of evolutionary game theory.

**Theorem 10** (Kwon and Mertikopoulos [2017], Theorem 4.1). *If $R$ is a regularizer function on $\mathcal{C}$ and $(\eta_t)_{t \in \mathbb{R}_+}$ is a positive, non-increasing and piecewise continuous parameter, then, for every locally intergrable payoff stream $(u_t)_{t \in \mathbb{R}_+} \in \mathbb{R}^d$, we have:*

$$\max_{x \in \mathcal{C}} \int_0^t \langle u_s, x \rangle ds - \int_0^t \langle u_s, x_s^c \rangle ds \leq \frac{R_{max} - R_{min}}{\eta_t}$$

## C   Constructing a DCFMM with AD securities via CFTRL

---

**Protocol 1** Constructing a DCFMM with AD securities via Continuous FTRL

---

**Given**: regularizer $R : \mathbb{R}^d \to \mathbb{R}, \nabla C = R, \mathcal{C} = \Delta_d$, and $t_0 = 0$.

*Repeat*:

1. Market receives trade bundle $r_t \in \mathbb{R}^d$

2. Charge trader for bundle $r_t$:

$$C(q_t + r_t) - C(q_t) = \int_0^1 \langle x_{t+1}^c(q_t + sr_t), r_t \rangle ds$$

where

$$x_{t+1}^c(q_t + sr_t) = \nabla C(q_t + sr_t) = \arg\max_{p \in \Delta_d} \langle p, \int_0^s q_t + tr_t dt \rangle - R(p)$$

which is CFTRL.

3. Set $q_{t+1} = q_t + r_t$ and t += 1

---

**Theorem 11.** *By CFTRL defined via Protocol 1 with a fixed learning rate of $\eta = 1$, the pricing of a bundle at a given state for a DCFMM can be expressed by an action taken by CFTRL. Thus, the worst-case loss of DCFMM is equivariant to the regret of CFTRL.*

*Proof.* At any round, the market receives a bundle $r_t \in \mathbb{R}^d$ and runs CFTRL

$$x_{t+1}^c(q_t + sr_t) = \nabla C(q_t + sr_t) = \arg\max_{p \in \Delta_d} \langle p, \int_0^s q_t + tr_t dt \rangle - R(p)$$

where the equality holds by Kwon and Mertikopoulos [2017][Proposition 2]. Also, observe that

$$\underbrace{\int_0^1 \langle x_{t+1}^c(q_t + sr_t), r_t \rangle ds}_{\text{parametric form}} = C(q_t + r_t) - C(q_t)$$

where the equality holds by the Fundamental Theorem of Line Integrals. Thus,

$$\underbrace{C(q_n) - C(q_0)}_{\text{Payments to Market}} = \sum_{t=0}^n C(q_t + r_t) - C(q_t) = \underbrace{\sum_{t=0}^n \int_0^1 \langle x_{t+1}^c(q_t + sr_t), r_t \rangle ds}_{\text{CFTRL Loss}}$$

and that

$$\underbrace{\max_{p \in \Delta_d} \langle p, q_n \rangle}_{\text{Worst-Case Market Payout}} = \max_{p \in \Delta_d} \langle p, q_0 + \sum_{t=0}^n r_t \rangle = \underbrace{\max_{p \in \Delta_d} \sum_{t=0}^n \int_0^1 \langle p, q_t + sr_t \rangle ds}_{\text{Worst-Case Action}}.$$

Hence, via the equalities, the regret rate of CFTRL matches the worst-case loss of the DCFMM. $\square$

**Corollary 1.** *Given a DCFMM defined via a strictly convex and differentiable $R$ defined over all of $\Delta_d$, then the worst-case loss is lower bounded by $R_{max} - R_{min} = \max_{p \in \rho(\mathcal{Y})} R(p) - \min_{p \in \Delta_d} R(p)$.*

Observe this worst-case loss matches the original derived worst-case loss in Abernethy et al. [2013][Theorem. 4.4.] for DCFMM with AD securities (up to a difference of max and sup in terms of $R$, we leave generalizing in terms of sup to future work).

# D   Gradient and Steepest Descent Results

**Theorem 12** (Ang [2025], Theorem 1). *If $f$ is convex and $\ell_2$-based $L$-smooth, then for gradient descent where $x_{t+1} = x_t - \gamma \nabla f(x_t)$, $\|x_{k+1} - x^*\|_2^2 < \|x_k - x^*\|_2^2$ if $x_k \notin \arg\min f$ and $\|x_{k+1} - x^*\|_2^2 = \|x_k - x^*\|_2^2$ otherwise.*

**Lemma 10** (Sidford [2024], Lemma 6.1.5). *For differentiable $f : \mathbb{R}^d \to \mathbb{R}$, norm $\|\cdot\|$, and $x \in \mathbb{R}^n$ define $Q : \mathbb{R}^n \to \mathbb{R}$ as*

$$Q(y) = f(x) + \langle \nabla f(x), y - x \rangle + \frac{L}{2}\|x - y\|^2$$

*where $\|\cdot\|$ is an arbitrary norm and $L > 0$. If $f$ is smooth w.r.t. $\|\cdot\|$ and $y^* = \arg\min_{y \in \mathbb{R}^d} Q(y)$ then $f(y^*) \leq f(x) - \frac{1}{2L}\|\nabla f(x)\|_*^2$.*

**Theorem 13** (Wright and Recht [2022], Theorem 3.5). *Suppose that $f$ is bounded below and is $L$-smooth w.r.t. $\|\cdot\|$. Then all accumulation points $\overline{x}$ of the sequence $\{x_t\}$ generated by a scheme that satisfies*

$$f(x_{t+1}) \leq f(x_t) - \frac{1}{2L}\|\nabla f(x_t)\|_*^2$$

*are stationary, that is, $\nabla f(\overline{x}) = 0$. If in addition $f$ is convex, each such $\overline{x}$ is a solution of $\min_{x \in \mathbb{R}^d} f(x)$.*

**Lemma 11** (Sidford [2024], Lemma 6.1.7). *If $f : \mathbb{R}^d \to \mathbb{R}$ is differentiable and $\mu$-strong convex w.r.t. $\|\cdot\|$ then for all $x_* \in \arg\min f$ and $x \in \mathbb{R}^n$ it holds that*

$$\frac{\mu}{2}\|x - x_*\|^2 \leq f(x) - f(x_*) \leq \frac{1}{2\mu}\|\nabla f(x)\|_*^2 \, .$$

**Lemma 12.** *If $f : \mathbb{R}^d \to \mathbb{R}$ is differentiable, $\mu$-strong convex w.r.t. $\|\cdot\|_\mu$, and $L$-smooth w.r.t. $\|\cdot\|_L$ then it holds that $\|y - x\|_\mu^2 \leq \frac{L}{\mu}\|x - x_*\|_L^2$ where $x_* \in \arg\min f$ and $x$ and $y$ are from Lemma 10.*

*Proof.* By Lemma 8, 10, and 11, we get

$$\frac{\mu}{2}\|y - x_*\|_\mu^2 \leq f(y) - f(x_*) \leq f(x) - f(x_*) \leq \frac{L}{2}\|x - x_*\|_L^2 \, .$$

Hence, we get $\|y - x\|_\mu^2 \leq \frac{L}{\mu}\|x - x_*\|_L^2$. $\qquad\square$

# E   Smooth Quadratic Prediction Markets with Constraints Analysis

**Buy-Only Market** For analysis sake, we substitute min argument of $r_t$ with $q_{t+1} = q_t + r_t$.

$$\min_{q_{t+1} \in \mathbb{R}^d} \quad \underbrace{\left( \langle \nabla C(q_t), q_{t+1} - q_t \rangle + \frac{L}{2}\|q_{t+1} - q_t\|_2^2 \right)}_{\text{Payment to Market}} - \underbrace{\langle \mu, q_{t+1} - q_t \rangle}_{\text{Expected Payout}}$$

$$\text{s.t.} \quad q_t \preceq q_{t+1}$$

Let $r = q_{t+1} - q_t$ and $c = \nabla C(q_t) - \mu$. This simplifies the objective too

$$\min_r \quad \langle c, r \rangle + \frac{L}{2}\|r\|_2^2$$

$$\text{s.t.} \quad -r \preceq 0.$$

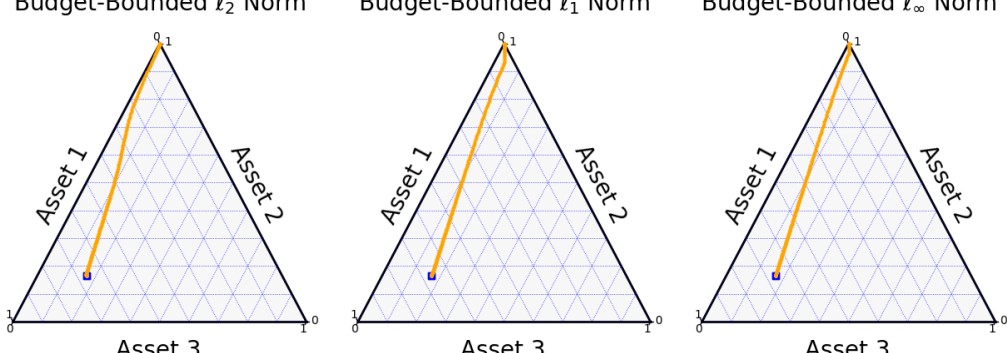

Figure 2: Let $q_0 = (10, 20, 10)$, $C$ is softmax with smoothness of $L = 1$, and $\mu = (1/6, 1/6, 2/3)$. The agents had a budget of $B = .01$. The blue square expresses $\mu$ and the orange path towards the blue square demonstrates the updating market distribution states. As denote by the titles's of each plot, we vary the norm used for the Smooth Quadratic Prediction Market. Note although softmax is not $\ell_1$-smooth, we use said norm experimentally for the sake of comparison.

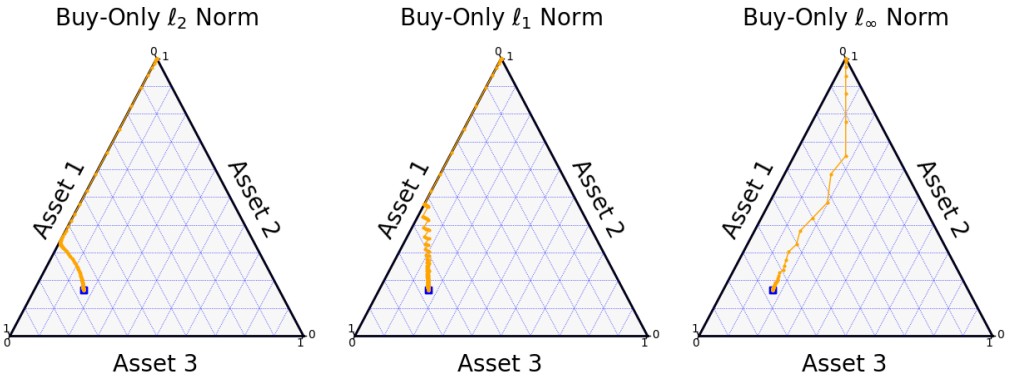

Figure 3: Let $q_0 = (10, 20, 10)$, $C$ is softmax with smoothness of $L = 1$, and $\mu = (1/6, 1/6, 2/3)$. The blue square expresses $\mu$ and the orange path towards the blue square demonstrates the updating market distribution states in a buy-only market. As denote by the titles's of each plot, we vary the norm used for the Smooth Quadratic Prediction Market. Note although softmax is not $\ell_1$-smooth, we use said norm experimentally for the sake of comparison.

Let the Lagrangian be denoted by $\mathcal{L}(r, \lambda) = \langle c, r \rangle + \|r\|_2^2 - \langle \lambda, r \rangle$ where $\lambda \in \mathbb{R}_+^d$. Solving for $r$ from $\min_{r \in \mathbb{R}^d} \mathcal{L}(r, \lambda)$, we get $r^* = \frac{1}{L}(\lambda - c)$. Note the dual problem is

$$\max_{\lambda \succeq 0} \frac{-1}{2L} \langle c, c \rangle - \frac{1}{2L} \langle \lambda, \lambda \rangle + \frac{1}{L} \langle \lambda, c \rangle .$$

We claim that $\lambda^* = c_+$. We verify $r^* = \frac{1}{L}(c_+ - c)$ and $\lambda^* = c_+$ via KKT conditions.

- **Stationarity** ($\frac{\partial \mathcal{L}}{\partial r_i^*}(r^*, \lambda^*) = 0 \ \forall \ i \in [d]$): $c + c_+ - c - c_+ = 0$

- **Complimentary Slackness** ($-\lambda_i^* r_i^* = 0 \ \forall \ i \in [d]$): If $c_i > 0$ then we have $\frac{-c_i}{L}(c_i - c_i) = 0$ and if $c_i \leq 0$ we have $\frac{-0}{L}(c_i - 0) = 0$.

- **Primal Feasibility**: If $c_i \geq 0$ then we have $\frac{1}{L}(c_i - c_i) = 0$ and if $c_i < 0$ we have $c_i < 0$.

- **Dual Feasibility**: $0 \preceq c_+$ by definition

Hence, we get that the update step is

$$q_{t+1} = q_t + \frac{1}{L}((\nabla C(q_t) - \mu)_+ - (\nabla C(q_t) - \mu))$$

and observe when $\nabla C(q_t) = \mu$ we have a stationary point.

## F  Smooth Quadratic Prediction Markets with Adaptive Liquidity

**Definition 15** (Asymmetric norm). *A function $g : \mathbb{R}^d \to \mathbb{R}_+$ is an asymmetric norm if it satisfies $\forall$ $x, y \in \mathbb{R}^d$:*

- *Non-negativity: $g(x) \geq 0$*

- *Definiteness: $g(x) = g(-x) = 0$ if and only if $x = 0$*

- *Positive homogeneity: $g(\alpha x) = \alpha g(x)$ for all $\alpha > 0$*

- *Triangle inequality: $g(x + y) \leq g(x) + g(y)$.*

**Lemma 9.** *$\text{Pay}_{L^\circ}$ satisfies Axiom 3 No Arbitrage.*

*Proof.* By (Lemma 3.6, Abernethy et al. [2014]) the VPM-DCFMM satisfies no arbitrage, i.e, for all $r_0, \ldots, r_t \in \mathbf{S}$ it holds that $\langle \rho(y), \sum_{i=0}^{t} r_i \rangle \leq \sum_{i=0}^{t} \text{Pay}_{D^\circ}(q_i, r_i; v_i)$ for some $y \in \mathcal{Y}$. However, it also holds that $\sum_{i=0}^{t} \text{Pay}_{D^\circ}(q_i, r_i; v_i) \leq \sum_{i=0}^{t} \text{Pay}_{L^\circ}(q_i, r_i; v_i)$. Therefore by combing the two inequalities we have that $\langle \rho(y), \sum_{i=0}^{t} r_i \rangle \leq \sum_{i=0}^{t} \text{Pay}_{L^\circ}(q_i, r_i; v_i)$. $\qquad\square$

