# OpenReview forum: "Smooth Quadratic Prediction Markets"
_NeurIPS.cc/2025/Conference — NeurIPS 2025 poster_

### Official Review · Reviewer_dHk7 · 2025-06-29

**Clarity:** 3
**Significance:** 3
**Originality:** 2
**Rating:** 5
**Confidence:** 3

**Summary:**

- The authors propose a novel prediction market model by replacing the fee of the traditional Duality-based Cost Function Market Maker (DCFMM) model (that is, the Bregman divergence between two consecutive states of the market) with a fee that is quadratic in the norm of the requested securities bundle *(Def. 5)*.
- The proposed novel model facilitates higher profits for the market maker, since the novel payment function upper bounds the traditional one *(Th. 4)*.
- The proposed model satisfies several of the desiderata for a prediction market, except incentive compatibility (which means that, given a market state and a trader's belief, the bundle requested by the trader in order to maximise the expected return shifts the market state to match the trader's belief). However, by showing that the update of the market state follows a gradient descent dynamic, it turns out that the novel model satisfies an *incremental* version of incentive compatibility (which means that an infinite sequence of return-maximiser trades with the same belief will shift the market state to asymptotically match the belief) *(Th. 8)*.
- Finally, incremental incentive compatibility (the convergence of the market state to the belief of a sequence of traders) is tested in two constrained scenarios (buy-only markets and markets with budget).

**Questions:**

### Major
- Incentive compatibility and its incremental version would benefit from a more extensive discussion and comparison. In particular, under incremental IC, the market state matches the traders' belief only asymptotically. What is the consequence of this in the short-run? What if traders with different beliefs trade one after the other? Can this lead to oscillatory market states?
- Beyond the definition of L-smoothness which allows for the bound of Theorem 4, what is the meaning / intuition of replacing the DCFMM Bregman fee with the quadratic one of Definition 5? Would other notions of distance be suitable - for example, the Bregman divergence of the strongly convex conjugate of $C$ between $C(q_t)$ and  $C(q_{t+1})$?
-  Can the bound of Theorem 4 be made tighter? How much better is the worst-case loss of SQPM over DCFMM?
- Given the relation between  DCFMM and FTRL, the relation between Smooth Quadratic Prediction Market and gradient descent upon the replacement of a Bregman term with a quadratic one is somewhat expected. What is the precise interplay between these elements?

### Minor
- Section 2.3 would benefit from a more explicit statement along the lines "If $\hat{C}$ (some property) then $C = \hat{C}^{\ast}$ is CIIP"
- L. 171: regrate is a typo
- Caption of Figure 1: the tiles in the plots do not seem to convey any information about the different norms
- Appendix F discusses Buy-Only Markets, but not Budget-Bounded Traders
- Line 411: $Q$ is the gradient of the convex conjugate of $R$, not the convex conjugate of $R$. Similarly, there is something wrong with the **Given** line of **Protocol 1** in Appendix D

**Ethical Concerns:**

["NO or VERY MINOR ethics concerns only"]

**Final Justification:**

The authors satisfactory addressed all of my questions. My rating remains unchanged.

**Limitations:**

- the authors acknowledge that their model does not fulfil one of the standard desirable axioms for prediction markets, and propose a sound alternative axiom which their model do fulfil
- an important  limitation might be that of agents with varying beliefs. It is not clear what happens of incremental incentive compatibility in this case. The authors acknowledge this limitation and propose it as a future direction.

**Paper Formatting Concerns:**

Conclusions should fit in the main body

**Quality:**

4

**Strengths And Weaknesses:**

### Strengths
- introduction and preliminaries are fairly clear and concise even for a reader not familiar with the literature
- the work is technically sound and supporting lemmas for each result are clearly stated, making the proofs self-contained and easier to follow
- the work is well written and organised into preliminaries, new model, its properties and limitations, experiments

### Weaknesses
- I am not familiar with the literature on prediction markets, but the generalisation over the traditional DCFMM model seems fairly mild, possibly hindering the impact and originality of the work

---

> ### Author Rebuttal · Authors · 2025-07-29
>
> > Incentive compatibility and its incremental version would benefit from a more extensive discussion and comparison. In particular, under incremental IC, the market state matches the traders' belief only asymptotically. What is the consequence of this in the short-run?
>
> Our reply: This is a key question for the proposed design. In terms of theory, we show that rates of convergence are fast. Additionally, since traders are incentivized to follow gradient descent, at least when the market uses the two-norm, we know that each trade moves the market state closer to a market state that maps to their belief (a non-asymptotic guarantee).
>
> In practice, if there is a constant fee or cost of effort to make each transaction, and/or if the liquidity of the market is high, we expect the trader to make few trades and stop quickly. So the behavior is expected to be very similar to the DCFMM in practice, but with a simpler and arguably preferable fee structure.
>
>
> > What if traders with different beliefs trade one after the other? Can this lead to oscillatory market states?
>
> Our reply: This is a bit unclear even in standard prediction markets. There are Bayesian models with information aggregation, but it’s not always clear how the theory of IC translates to actual markets with many participants. However, these markets do work well and converge in practice, and the SQPM should be no different
>
>
> > Beyond the definition of L-smoothness which allows for the bound of Theorem 4, what is the meaning / intuition of replacing the DCFMM Bregman fee with the quadratic one of Definition 5? Would other notions of distance be suitable - for example, the Bregman divergence of the strongly convex conjugate of $C$ between $C(q_t)$ and $C(q_{t+1})$?
>
> Our reply: It’s actually quite tricky, we think experts would be surprised that it is possible at all to find a fee that threads the needle between:
> large enough to maintain the market maker’s worst-case loss, but
> small enough to retain a version of incentive compatibility: the agents aren’t charged too much, so they push the prices to match their beliefs.
> We don’t know of another fee proposal that would work. However, the intuition for the quadratic fee can be viewed from several angles: It uses the quadratic upper-bound that is tangent to the cost function at the current point, so it is “locally” curved correctly. This technique in optimization is called majorization-minimization, which gives another intuition.
>
> > Can the bound of Theorem 4 be made tighter? How much better is the worst-case loss of SQPM over DCFMM?
>
>
> Our reply: We believe that the difference can be made arbitrarily small: take any sequence of trades, and “split” the trades to be a large number of arbitrarily small trades. In this case, the worst-case loss of the two designs should converge to be the same.
>
>
> > Given the relation between DCFMM and FTRL, the relation between Smooth Quadratic Prediction Market and gradient descent upon the replacement of a Bregman term with a quadratic one is somewhat expected. What is the precise interplay between these elements?
>
> Our reply: While the relationship to FTRL was noted, the literature didn’t seem to know how to exploit it previously – even in this work, our expansion to continuous FTRL in the appendix is novel. Additionally, our presentation of the DCFMM as a breakdown into a linear term plus a Bregman term is not the standard presentation, and part of our contribution is noting that this breakdown is useful and can be modified to obtain the quadratic fee market. The relation to gradient descent: One thing that is unexpected is that the market maker is still using FTRL as its algorithm to update prices. Yet the strategic process surprisingly results in gradient descent on a related function, the convex conjugate of the FTRL regularizer.
>
> > Minor questions:
>
> Our reply: thank you for the feedback, we acknowledge the presented typos and unclear aspects and will use this feedback to improve the writing.

---

> > ### Comment · Reviewer_dHk7 · 2025-08-02
> >
> > Thanks for addressing my questions; in particular, I find interesting the non-asymptotic guarantee of the first point. My rating remains unchanged.

---

### Official Review · Reviewer_UR8r · 2025-07-02

**Clarity:** 3
**Significance:** 2
**Originality:** 3
**Rating:** 4
**Confidence:** 3

**Summary:**

This paper studies the prediction market design problem. The authors first introduced the classical Duality-based Cost Function Market Maker (DCFMM) and its connection to Follow-The-Regularized-Leader (FTRL). By decomposing and modifying the DCFMM's pricing mechanism, this paper introduces the Smooth Quadratic Prediction Market, a novel prediction market design inspired by general steepest gradient descent. Specifically for Arrow-Debreu (AD) securities, the authors show Smooth Quadratic Prediction Market offers a better worst-case loss while preserving key axiomatic guarantees. The paper further explores agent behavior under realistic constraints (bounded budgets and buy-only securities) and provides an introductory analysis of adaptive liquidity.

**Questions:**

Could you elaborate on the practical implications of only satisfying "incremental incentive compatibility"? Are there scenarios where this weaker guarantee could lead to undesirable market behaviors?

**Ethical Concerns:**

["NO or VERY MINOR ethics concerns only"]

**Final Justification:**

The authors addressed my questions, I raised my score.

**Limitations:**

yes

**Quality:**

2

**Strengths And Weaknesses:**

On the positive side, this paper considers a different approach for prediction market design, exploring a new direction for prediction market design beyond the well-established DCFMM framework. I found the problem presented by the paper interesting and relevant. In general, this paper proposed an innovative research topic.

On the other hand, while the paper acknowledges that the Smooth Quadratic Prediction Market does not satisfy the strong axiom of incentive compatibility, instead relying on an "incremental incentive compatibility," the implications of this weaker form for real-world market behavior and equilibrium outcomes could be discussed in more detail. This might be a critical point for practical adoption. For example, the definition requires “a sequence of agents with the same belief”; is it reasonable to assume that there exists a sequence of agents that will have exactly the same belief in practice?

While the paper provides an analytical proof (Theorem 4) that the Smooth Quadratic Prediction Market has a better worst-case loss than the DCFMM, *empirical validation* would significantly strengthen this claim.

In Section 5, the acknowledgment that proving other important properties (bounded-worst case loss, information incorporation, incentive compatibility) for the adaptive liquidity approach becomes challenging due to non-convexity is honest but highlights a significant limitation for this promising extension. The section feels somewhat incomplete due to these open problems.

---

> ### Author Rebuttal · Authors · 2025-07-29
>
> > Could you elaborate on the practical implications of only satisfying "incremental incentive compatibility"? Are there scenarios where this weaker guarantee could lead to undesirable market behaviors?
>
> Our reply: Due to the similarity of questions, we refer our response to the reply of 7cVQ’s first question above.
>
> > Is it reasonable to assume that there exists a sequence of agents that will have exactly the same belief in practice?
>
> Our reply: We are not necessarily stating that we expect a sequence of agents with the same beliefs to come to the market; however, we primarily aim to demonstrate that the market functions as a belief aggregator in this particular scenario (a common baseline expectation in the prediction market literature). Realistically, each trader will have a unique belief. Even so, given that traders are incentivized to follow gradient descent (at least when the market uses the two norm), each trader's purchase moves the market state towards a state that maps to their particular belief (this is a byproduct of Theorem 12 in the appendix).

---

> > ### Comment · Reviewer_UR8r · 2025-08-06
> >
> > Thank you for your responses. I will take them into account during the AC-Reviewer discussion phase. I don’t have any further questions.

---

### Official Review · Reviewer_7cVQ · 2025-07-03

**Clarity:** 4
**Significance:** 3
**Originality:** 3
**Rating:** 5
**Confidence:** 2

**Summary:**

The paper proposes Smooth Quadratic Prediction Markets (SQPM), a variant of Duality-based Cost Function Market Maker (DCFMM) for Arrow-Debereu securities. By replacing the DCFMM’s Bregman “curvature fee” with a simple quadratic fee derived from an L-smooth upper bound, the SQPM retains axioms of automated market makers such as instantaneous price, information incorporation and no arbitrage. Moreover, their SQPM incentivizes traders to implement general steepest descent paths toward their beliefs, satisfying incremental incentive compatibility. Lastly, they suggest an extension to adaptive liquidity by allowing smoothness to decrease as trading volume grows.

**Questions:**

- In the absence of one‐step incentive compatibility, could a trader exploit incremental steps (e.g., splitting a large trade)?
- How should one choose the norm ∥·∥ and smoothness parameter L in real‐world settings?

**Ethical Concerns:**

["NO or VERY MINOR ethics concerns only"]

**Final Justification:**

Although, it is not exactly my area of expertise, I remain confident about the contributions of this paper.

**Limitations:**

yes

**Paper Formatting Concerns:**

There are no formatting issues in this paper.

**Quality:**

4

**Strengths And Weaknesses:**

### Strenghts
- The paper seems to make novel conncection to optimization by extending the duality between FTRL to general L-smooth cost functions.
- The paper is well written; the theorems are clearly stated and proved, to the degree that I can understand.

---

> ### Author Rebuttal · Authors · 2025-07-29
>
> > In the absence of one‐step incentive compatibility, could a trader exploit incremental steps (e.g., splitting a large trade)?
>
> Our reply: The agent will prefer to split a large trade over several trades rather than making a single large trade. This is a potential drawback, but we view it as benign for several reasons:
> There’s no arbitrage opportunity or free profit available; the market maker’s loss is still bounded. Hence, we would not view trade-splitting as an “exploit” opportunity.
> If the market charges a constant fee to transact (or we imagine a small cost of doing each transaction due to friction), the trader will not want to split trades below a certain level. We leave formalization of this to future work.
> If trades are small relative to the size of the market (i.e., the liquidity), the benefit from splitting trades is extremely small. So for typical traders, this will not apply. On the other hand, for extremely large trades, it might even be preferable to the market designer if the trader splits the trades, which gives a more continuous price path.
>
>
> > How should one choose the norm ∥·∥ and smoothness parameter L in real‐world settings?
>
> Our reply:
>
> We clarify that the designer needs to choose the convex cost function $C$ and the norm, after which L should always be chosen to equal the smooth convexity parameter of $C$ with respect to that norm.
>
> Choosing $C$ is a significant question in the prediction market literature, with some understanding but no simple answer. We can “stretch” $C$ by scaling its convex conjugate such that L is scaled up or down. In general, stretching to obtain a larger L (more curvature) will result in a better worst-case loss, but will provide less liquidity to the market because prices change more rapidly in response to trades. A smaller L will have a large worst-case loss, but it provides a lot of liquidity.
>
> Regarding the choice of norm, one impact we have observed is on the dynamics of prices. The way trades evolve for L1 versus L2 is different. L1 and L infinity have a “sparse” behavior compared to L2. A trade in one asset is more likely to affect only a few prices, rather than all prices. But it changes those few prices more dramatically, so the updates are less smooth. (See figure 1 for an illustration.)

---

### Official Review · Reviewer_gCrc · 2025-07-03

**Clarity:** 3
**Significance:** 3
**Originality:** 3
**Rating:** 5
**Confidence:** 1

**Summary:**

The paper studies automated market makers for Arrow–Debreu securities.
The market-maker maintains a state vector $q_t\in\mathbb{R}^d$ recording the total shares held by traders.
When a trader requests a bundle $r_t\in\mathbb{R}^d$ (positive to buy, negative to sell), the market maker charges $\textrm{pay}(q_t,r_t)$ and updates the state to $q_{t+1}:=q_t+r_t$. After trading ends a single outcome $Y\in[d]$ is revealed and, for any $t$, the $t$-th trader earns $r_t(Y)$.
In the classical dual-cost function market maker (DCFMM) one assumes a convex, increasing, 1-invariant, probability-mapping  (CIIP) potential $C:\mathbb{R}^d\to\mathbb{R}$ that is the Fenchel conjugate of some strictly convex and continuous $\hat{C}$ defined in the relative interior of the probability simplex $\Delta_{d}$ and sets  $\text{pay}(q,r):=C(q+r)-C(q)$.
This choice satisfies axioms 1–5 (price existence, information incorporation, no-arbitrage, expressiveness, one-shot incentive compatibility) and gives a finite worst-case loss.
The paper adds the assumption that $C$ is $L$-smooth, so that $C(q+r)\le C(q)+\langle\nabla C(q),r\rangle+\tfrac{L}{2}\|r\|^{2}$ and proposes charging this upper bound instead, i.e., $\textrm{pay}(q,r):=\langle\nabla C(q),r\rangle+\tfrac{L}{2}\|r\|^{2}.$
The new rule (SQPM) preserves axioms 1–4, lowers the market maker’s worst-case loss, and replaces Axiom 5 with a weaker Axiom 6: incremental incentive compatibility.
A risk-neutral trader’s best response is a steepest-descent step toward her belief, and repeated budget-limited trades converge to that belief at rate $O(1/t)$.
The authors also discuss budget constraints, buy-only settings, and a sketch for adaptive liquidity.

**Questions:**

Typos:
- line 67 (missing f)
- line 101 (instsantaneous)
- line 127 (missing full stop)
- line 171 (regrate)

**Ethical Concerns:**

["NO or VERY MINOR ethics concerns only"]

**Final Justification:**

I'm not familiar with the line of work this paper is about, but I enjoyed reading it and to me it's a valuable contribution. Hence, even if with low confidence, I recommend the paper for acceptance.

**Limitations:**

yes

**Quality:**

3

**Strengths And Weaknesses:**

The upper bound idea is nice, the paper seems well-written, and the statements well-supported by theorems and proofs.

---

### Decision · Program_Chairs · 2025-09-17

**Decision:**

Accept (poster)

**Comment:**

The reviewers all agreed that this paper provided interesting technical contributions to the field of prediction markets, with novel connections to optimizations and online learning. The reviewers also agreed that this paper was well-written, and the main results were clearly presented. The reviewers were largely positive before the rebuttal phase, and feel that the authors satisfactorily addressed any remaining concerns through their rebuttals.